# Neurological manifestations and *MMP8* as a prognostic biomarker in severe fever with thrombocytopenia syndrome

Qi Xia[1☿], Ziling Cheng[1☿], Yi Zhang[2], Haolin Song[1], Bei Jia[3], Lingtong Huang[4], Qiuhong Liu[1], Qing Zhao[1], Jie Li[3]*, Jie Wang[1]*, Wei Wu[1]*

1 State Key Laboratory for Diagnosis and Treatment of Infectious Diseases, The First Affiliated Hospital, Zhejiang University School of Medicine, Hangzhou, China, 2 Department of Laboratory Medicine, The First Affiliated Hospital, Zhejiang University School of Medicine, Hangzhou, China, 3 Department of Infectious Diseases, Nanjing Drum Tower Hospital, Affiliated Hospital of Medical School, Nanjing University, Nanjing, Jiangsu, China, 4 Department of Critical Care Units, The First Affiliated Hospital, Zhejiang University School of Medicine, Hangzhou, China

☿ These authors contributed equally to this work.
* lijier@nju.edu.cn (JL); wangjie_lethe@zju.edu.cn (JW); zjuwuwei@zju.edu.cn (WW)

## Abstract

### Background

Severe fever with thrombocytopenia syndrome (SFTS) is an emerging infectious disease with high mortality. Neurological symptoms are increasingly recognized as critical predictors of poor outcomevs, but their mechanisms remain unclear. This study aims to explore relevant mechanisms and identify biomarkers.

### Methods

This study utilized three distinct SFTS patient cohorts. First, 15 hospitalized patients from April 21 to August 2, 2024 were prospectively enrolled, with blood samples collected during hospitalization for RNA sequencing of peripheral blood mononuclear cells (PBMCs) to identify differentially expressed genes (DEGs). Second, we established a single-center retrospective cohort comprising 103 SFTS patients (admitted 2021–2024) to analyze clinical data pertaining to central nervous system (CNS) symptoms and patient outcomes. Third, to quantitatively assess serum matrix metalloproteinase-8 (*MMP8*) levels, we analyzed a total of 151 serum samples obtained from patients across two independent centers using enzyme-linked immunosorbent assay (ELISA).

### Results

RNA sequencing analysis between the recovered group and deceased group revealed significant alterations in central nervous system (CNS)-related pathways. Clinical data from 103 SFTS patients showed the deceased group had significantly

**Data availability statement:** The raw sequence data reported in this paper have been deposited in the Genome Sequence Archive (Genomics, Proteomics & Bioinformatics 2021) in National Genomics Data Center (Nucleic Acids Res 2024), China National Center for Bioinformation / Beijing Institute of Genomics, Chinese Academy of Sciences (GSA-Human: HRA010122) that are publicly accessible at https://ngdc.cncb.ac.cn/gsa-human.

**Funding:** This work was supported by the National Key Research and Development Program of China (No.2023YFC2506000 to WW and 2021YFC2301800 to ZY) and Natural Science Foundation of China (NSFC81900572 to WJ). The funders had no role in study design, data collection and analysis, decision to publish, or preparation of the manuscript.

**Competing interests:** The authors have declared that no competing interests exist.

higher rates of neurological symptoms vs the recovered group over the full course: consciousness disorders (88.89% vs 23.53%, P<0.001) and convulsions (27.78% vs 3.53%, P=0.003). Patients with these symptoms had both upregulated *MMP8* gene expression and elevated serum *MMP8* levels. Elevated serum *MMP8* strongly correlated with fatal outcomes (AUC=0.821, 95% CI: 0.681-0.962; P=0.008). Multivariate Cox analysis confirmed *MMP8* as an independent mortality predictor (HR=1.060, 95% CI: 1.010-1.112). The *MMP8*-incorporated multivariable Cox model showed good discriminative ability in the external validation cohort (AUC=0.843, 95% CI: 0.723-0.962).

## Conclusions

Neurological symptoms during the early stages of SFTS are strongly associated with patient outcomes. *MMP8* may serve as a potential biomarker for SFTS prognosis. The integration of clinical neurological symptoms and *MMP8* measurement offers a novel framework for improving prognostic accuracy and guiding personalized management strategies in SFTS patients.

### Author summary

This study investigates the relationship between severe fever with thrombocytopenia syndrome (SFTS) patients' outcome, neurological symptoms, and transcriptome changes, addressing critical knowledge gaps in this emerging infectious disease. Patients were categorized into deceased group and recovered group based on their clinical outcomes. By comparing gene expression profiles and clinical neurological symptoms between these groups, we identified *matrix metalloproteinase-8 (MMP8)* as a potential biomarker for predicting poor prognosis in SFTS patients. Furthermore, our results indicated that among the various neurological manifestations in SFTS patients, the presence of consciousness disorder and convulsion may serve as markers of poor prognoses. These findings advance the understanding of the neurological and molecular factors underpinning SFTS. We aim to assist clinicians in identifying SFTS patients with poor prognosis early, and providing timely supportive treatment for these high-risk patients to improve their survival outcomes.

## Introduction

Severe Fever with Thrombocytopenia Syndrome (SFTS) is an emerging tick-borne infectious disease caused by the Dabie bandavirus (DBV), also known as the SFTS virus (SFTSV). It is classified in the Banyangvirus genus, within the Phenuiviridae family and the Bunyavirales order [1,2]. Since its identification in China in 2009, SFTS has become a major public health concern due to its high mortality rate, which

**PLOS** **Neglected Tropical Diseases**

ranges from 5% to 30% [2–4]. The disease is now prevalent in several East Asian countries, including China, Japan, South Korea, and Vietnam [5], where it presents a seasonal threat, particularly to older adults and individuals with pre-existing conditions [6–8]. Furthermore, recent serological evidence suggests its geographic scope is expanding, with related viruses also being reported in Pakistan [9] and Kenya [10].

The course of SFTS typically progresses through four stages: incubation, fever, organ failure, and either recovery or death. The incubation period after SFTSV infection typically ranges from 5 to 14 days. The acute phase is characterized by high fever (lasting for about 7 days), thrombocytopenia, leukopenia, and elevated liver enzymes [11]. Severe cases can manifest as hemorrhage, neurological damage, myocardial damage, multiple organ dysfunction, and even death [12–14]. *Wang et al.* categorized the progression of SFTS into four stages based on clinical manifestations: S1 (0–7 days, Fever stage), S2 (8–14 days, Deterioration/Organ failure), S3 (15–21 days, Improving), S4 (22 days and beyond, Convalescence). Ribavirin is widely utilized in clinical practice for antiviral treatment of SFTS patients, as recommended by the Chinese management guidelines for SFTS issued in 2010, however, its efficacy remains uncertain [15–17]. Due to the absence of approved vaccines and effective antiviral drugs for SFTS, the mortality rate of SFTS has remained high over the years [18]. Consequently, timely diagnosis and appropriate supportive care remain the most effective strategies for managing SFTS cases [19].

While SFTS primarily affects the hematological and hepatic systems, neurological symptoms are increasingly recognized as critical factors associated with poor outcomes [20,21]. Neurological symptoms, platelet count, aspartate aminotransferase (AST) level, and CRP levels have been considered to be significant predictors of critical illness in SFTS patients [22,23]. Despite their clinical significance, the molecular basis of neurological symptoms remains poorly understood.

In this study, by collecting specific neurological symptoms manifested throughout the disease course and specifically during the early disease stage (S1 phase) in SFTS patients, we aim to identify early-stage neurological symptoms with predictive prognostic value. Additionally, RNA transcriptome analysis of peripheral blood mononuclear cells (PBMCs) from SFTS patients was performed to characterize differential gene expression between fatal cases and recovered patients, as well as between patients presenting with and without neurological symptoms.

Finally, the transcriptome-derived prognostic indicators were validated in serum samples from two independent SFTS patient cohorts, thereby enabling the identification of high-risk patients through a sensitive, accurate, and practical predictive tool and contributing to enhanced clinical decision-making and personalized treatment strategies.

## Methods

### Ethics statement

This study was approved by the Clinical Research Ethics Committee of the First Affiliated Hospital, College of Medicine, Zhejiang University (approval number 2022–119) and complied with the principles of the Declaration of Helsinki. Written informed consent was obtained from all patients or their immediate relatives for sample and data collection, as well as their use in research.

**Participants and samples** 15 patients infected with SFTSV were recruited from Center A (The First Affiliated Hospital of Medical School of Zhejiang University) between April 21 and August 2, 2024. We categorized the progression of SFTS into four stages based on clinical manifestations: S1 (0–7 days, Fever stage), S2 (8–14 days, Deterioration/Organ failure), S3 (15–21 days, Improving), S4 (22 days and beyond, Convalescence). Based on the progression of the disease, some patients underwent multiple sample collection at different stages of SFTS. An initial blood sample was collected from all 15 patients within 24 hours of hospital admission. To capture dynamic gene expression changes, longitudinal samples were subsequently collected from a subset of these patients (3 recovered and 1 deceased) at different stages of the disease (S1, S2, and S3). In total, 23 PBMC samples were collected from 15 SFTS patients and used for RNA-seq analysis.

Furthermore, the Validation Cohort 1 (n = 103; retrospective patients from Center A admitted between July, 2021-May, 2024) was analyzed to investigate neurological symptom-prognosis correlations. Serum *MMP8* concentrations were quantified in Validation Cohort 2A (43 patients from Center A) and Validation Cohort 2B (108 patients from Center B: Nanjing Drum Tower Hospital) to verify biomarker performance. Additionally, a cohort of 19 age- and sex-matched healthy individuals was recruited from Center A to serve as a healthy control group for serum *MMP8* level comparison. Clinical information was collected from the medical records of the patients following a standard protocol during their clinical course. Baseline laboratory data were derived from fresh blood samples collected at the time of sampling. In accordance with the Chinese management guidelines for SFTS (2023), patients received antiviral therapy (ribavirin) and supportive treatments. The clinical outcomes (recovered or deceased) were followed up until November 30, 2024.

**Diagnostic criteria** All patients were diagnosed based on SFTS criteria and confirmed via SFTSV nucleic acid testing. According to China's 2023 Guidelines for Diagnosis and Management of Severe Fever with Thrombocytopenia Syndrome (SFTS), suspected cases are defined by exposure history alongside characteristic clinical presentations (e.g., fever, fatigue, nausea/vomiting). Confirmation requires meeting suspected case criteria plus ≥1 laboratory finding: (1) SFTSV nucleic acid positivity; (2) Viral isolation from plasma; (3) SFTSV-IgG seroconversion or ≥4-fold titer rise in convalescent versus acute-phase sera.

**Clinical Assessment of Neurological Symptoms** Neurological symptoms were systematically recorded throughout the clinical course. Consciousness Disorder: This was defined as a moderate-to-severe impairment of consciousness, corresponding to a Glasgow Coma Scale (GCS) score of 12 or less. Restlessness: This was defined as agitation corresponding to a Richmond Agitation-Sedation Scale (RASS) score of +1. Mental State Change: This was primarily defined as apathy, characterized by a diminished motivation and reduced spontaneous interaction with the surroundings.

**SFTSV RNA detection** The presence of the SFTS virus in patients was determined using the SFTS Nucleic Acid Detection Kit, DaAn Gene (Guangzhou, China). The kit utilizes a one-step real-time reverse transcription polymerase chain reaction(RT-PCR) to design specific primers and fluorescent probes targeting the highly conserved region of the SFTSV gene coding region for RT-PCR amplification to quantitatively detect SFTSV RNA in serum specimens [24].

**RNA extraction, Library construction, and Sequencing** Total RNA was extracted using Trizol reagent (Thermo Fisher, 15596018) following the manufacturer's procedure. The quantity and purity of total RNA were analyzed by Bioanalyzer 2100 and RNA 6000 Nano Kit (Agilent, USA), and high-quality RNA samples with RIN number>7.0 were used to construct a sequencing library. The sequencing library was used for transcriptome sequencing, which was performed using the Illumina NovaSeq 6000 (LC-Bio Technologies, Hangzhou, China).

**Differential Gene Expression and Enrichment Analysis** Differentially expressed genes (DEGs) were identified using DESeq2 software (http://www.bioconductor.org/packages/release/bioc/html/DESeq2.html). Genes with a p-value <0.05 and an absolute fold change ≥2 were considered DEGs and subjected to Gene Ontology (GO) enrichment analysis using the GO database (http://www.geneontology.org/).

**Measurement of Serum *MMP8* Levels** Serum matrix metalloproteinase 8 (*MMP8*) concentrations were measured using a commercial human *MMP8* enzyme-linked immunosorbent assay (ELISA) kit (*Cat. No. EK1M08; LiankeBio, Hangzhou, China*), according to the manufacturer's instructions.

**Statistical analysis** The SPSS software, version 27.0 (IBM Corp., Armonk, NY, USA), R (version 4.4.2), and GraphPad Prism (version 10.1.2, San Diego, CA, USA) were used to analyze the statistical data. Categorical variables are presented as frequency counts and percentages [n (%)]. Intergroup comparisons for categorical data were conducted using the $\chi^2$ test or Fisher's exact test, as appropriate. Normally distributed continuous variables are expressed as mean ± standard deviation (mean ± SD) and compared using independent samples t-tests. Non-normally distributed continuous variables are reported as median with interquartile range [M (P25-P75)]; comparisons between groups utilized the Mann-Whitney *U* test. For comparisons involving three or more groups, the Kruskal-Wallis test followed by Dunn's post-hoc test for multiple comparisons was used. Univariate analyses employed $\chi^2$ tests or Fisher's exact tests. Multivariate analysis was performed

using Cox proportional hazards regression models, with candidate variables selected via a forward stepwise method (entry P < 0.10). All reported p-values are two-sided, with statistical significance set at α = 0.05.

## Results

### Baseline characteristics of SFTS patients

A total of 15 SFTS patients were included, with 11 patients (3 in S1, 8 in S2-4) in the recovered group and 4 patients (3 in S1, 1 in S2-4) in the deceased group. Patients were enrolled between April 21 and August 2, 2024, aligning with the typical seasonal occurrence of SFTS (Fig 1). The mean age of the patients was 70.0 years (interquartile range[IQR]:68.0-75.0). Patients in the deceased group were significantly older than those in the recovered group (76.5 ± 6.6 years vs 67.7 ± 6.9 years; P = 0.046) (Table 1). Aligning with prevailing evidence, our analysis demonstrated significant differences in age and key pathophysiological indicators—specifically coagulation parameters, hepatic/renal biomarkers, and tissue damage markers—between mortality and survival groups. Coagulation profiles revealed substantial differences, with deceased patients exhibiting significantly prolonged APTT values compared to recovered patients (P < 0.001). Additionally, deceased patients exhibited significantly elevated levels of D-dimer (median: 4.68 mg/L FEU vs 1.11 mg/L FEU, P = 0.026), ALT (median: 104.0 U/L vs 66.0 U/L, P = 0.028), AST (median: 357.0 U/L vs 67.0 U/L, P = 0.026), LDH (median: 798.5 U/L vs 317.0U/L, P = 0.026), and CK (median: 2351.0 U/L vs 76.0 U/L, P = 0.006), indicating impaired liver function and muscle injury in deceased patients.

### Neurological pathway enrichment with prognostic implication in SFTS patients' PBMCs

To explore the differences in gene expression between SFTS patients who recovered and those who died, we performed RNA-seq analysis on samples from 4 deceased and 11 recovered patients. Differentially expressed genes (DEGs) were

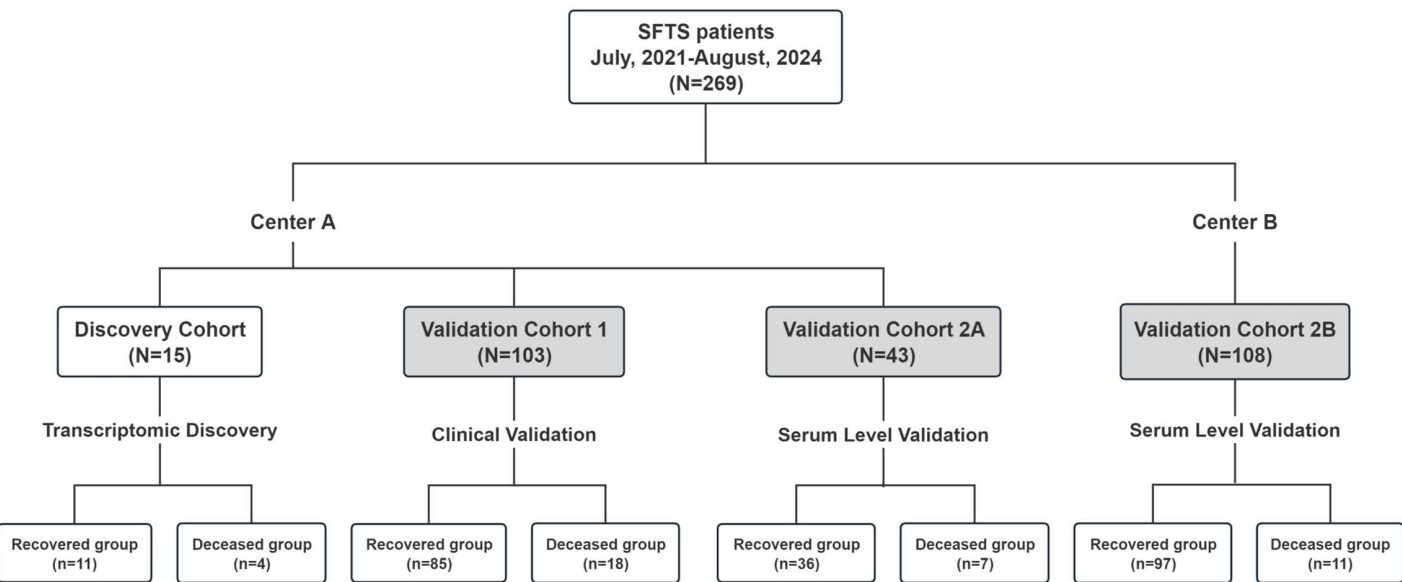

**Fig 1. Study Design Schema: Multi-Cohort Patient Allocation for Transcriptomic Discovery and Serum Biomarker Validation in SFTS Prognosis. Center A** (the First Affiliated Hospital, College of Medicine, Zhejiang University): Discovery Cohort (n = 15) for transcriptomic analysis; Validation Cohort 1 (n = 103) for clinical correlation; Validation Cohort 2A (n = 43) for initial serum biomarker testing. **Center B** (Nanjing Drum Tower Hospital): Validation Cohort 2B (n = 108) for external serum verification. Differential analysis was performed according to the final outcome (recovery/ death) of patients in all validation cohorts.

**Table 1. Baseline Characteristics of SFTS Patients.**

| Variables | Total | | | | S1 (Stage 1) | | |
|---|---|---|---|---|---|---|---|
| | Recovered (n=11) | Deceased (n=4) | *P* value | Recovered (n=3) | Deceased (n=3) | *P* value |
| *Demographics* | | | | | | | |
| Age, years | 67.70±6.90 | 76.50±6.60 | **0.046** | 71.00±3.50 | 81.70±1.20 | **0.007** |
| Males,n (%) | 5 (45.45) | 3 (75.00) | 0.569 | 1 (33.33) | 1 (33.33) | 0.414 |
| Hypertension,n (%) | 7 (63.64) | 3 (75.00) | 0.569 | 3 (100.00) | 2 (66.67) | 0.208 |
| Diabetes, n (%) | 2 (18.18) | 0 (0.00) | 1.000 | 2 (66.67) | 0(0.00) | 0.386 |
| Tumor, n (%) | 2 (18.18) | 2 (50.00) | 0.516 | 2 (66.67) | 1(33.33) | 0.410 |
| *Laboratory parameters* | | | | | | | |
| WBC,×$10^9$/L | 4.90±3.23 | 6.10±4.29 | 0.568 | 4.11±2.44 | 4.59±1.87 | 0.825 |
| Neutrophil (%) | 64.70(47.90-80.80) | 88.20(61.70-91.30) | 0.104 | 72.60±11.80 | 79.00±22.10 | 0.507 |
| Lymphocyte(%) | 26.70(11.60-43.20) | 8.65(7.13-24.65) | 0.078 | 21.20±13.30 | 14.50±13.30 | 0.268 |
| RBC,×$10^{12}$/L | 4.26±0.64 | 4.51±0.97 | 0.560 | 4.74±0.65 | 3.90±0.13 | **0.046** |
| Hemoglobin, g/L | 129.60±19.60 | 139.80±29.40 | 0.451 | 141.00±19.10 | 121.00±10.40 | 0.268 |
| Platelet,×$10^9$/L | 73.00 (42.00-128.00) | 47.50 (29.30-77.80) | 0.280 | 62.30±19.60 | 64.30±30.60 | 0.507 |
| CRP, mg/L | 4.27(1.00-7.92) | 13.11(4.52-51.10) | 0.177 | 5.13±2.51 | 22.84±34.31 | 0.507 |
| PT, s | 11.20±0.66 | 11.03±0.93 | 0.690 | 11.40±0.98 | 10.90±1.04 | 0.268 |
| APTT, s | 33.43±6.72 | 59.70±16.71 | **<0.001** | 38.20±6.87 | 45.00±11.43 | 0.268 |
| INR | 0.98±0.07 | 0.97±0.08 | 0.747 | 1.00±0.09 | 0.96±0.10 | 0.268 |
| D-dimer, mg/L FEU | 1.11 (0.66-3.79) | 4.68 (3.12-8.81) | **0.026** | 0.88 (0.66-3.79) | 5.30 (2.81-5.30) | 0.121 |
| ALB, g/L | 34.04±3.01 | 30.13±3.96 | 0.060 | 35.03±4.18 | 32.70±4.85 | 0.825 |
| ALT, U/L | 66.00 (26.00-110.00) | 104.00 (44.50-318.00) | **0.028** | 69.00±37.70 | 45.00±3.50 | 0.507 |
| AST, U/L | 67.0(38.0-138.0) | 357.0(151.5-1335.8) | 0.026 | 117.0±85.0 | 162.70±77.40 | 0.268 |
| TBIL, μmol/L | 11.54±7.93 | 11.23±8.01 | 0.948 | 5.80±3.10 | 8.93±9.24 | 0.825 |
| sCr, μmol/L | 66.00±15.70 | 134.50±61.00 | 0.109 | 71.00±26.00 | 130.30±76.80 | 0.268 |
| BUN, mmol/L | 6.29±2.90 | 15.54±7.35 | 0.083 | 4.59±2.15 | 13.98±9.20 | **0.046** |
| LDH, U/L | 317.00 (257.00-522.00) | 798.50 (470.80-1599.50) | **0.026** | 530.70±328.10 | 491.30±142.60 | 0.825 |
| CK, U/L | 76.00 (42.00-161.00) | 2351.00 (545.50-4092.00) | **0.006** | 237.00±152.90 | 1213.30±1713.60 | 0.268 |
| Viral Load, $TCID_{50}$/ml | 238.47 (9.00-5323.00) | 3243434.38 (1635919.69-4989780.44) | **0.001** | | | |

Data are median (IQR) or n (%) or mean±standard deviation. *P* values were calculated by two-sided Mann-Whitney U test, t test or Pearson's chi-squared test according to the normal distribution of data. S1:0–7 days, Fever stage. WBC white blood cell, RBC red blood cell, CRP C-reactive protein, PT prothrombin time, APTT activated partial thromboplastin time, ALT alanine aminotransferase, AST aspartate aminotransferase, TBIL total bilirubin, sCr serum creatinine, BUN blood urea nitrogen, LDH lactate dehydrogenase, CK creatine kinase.

identified from the RNA-seq data of all samples. Hierarchical clustering of transcriptome-wide gene expression (Fig 2A) revealed that samples from deceased patients formed a distinct cluster, separate from recovered patients, indicating significant systemic differences in RNA expression. Differential gene expression analysis identified 1924 DEGs (P<0.05, exact test, likelihood ratio test, and quasi-likelihood F test), with 479 genes downregulated and 1445 genes upregulated in the deceased group compared to the recovered group (Fig 2B). Gene Ontology (GO) enrichment analysis using clusterProfiler revealed that pathways associated with nervous system development were significantly enriched in deceased patients (Fig 2C, 2F). Given the small sample size, we also performed a more stringent analysis using an FDR-adjusted p-value (q-value<0.05) (S1 Fig), which identified 1693 DEGs (611 upregulated, 1082 downregulated). Reassuringly, GO analysis of this stricter gene set also revealed a significant enrichment of the nervous system development pathway. This finding was consistent with prior studies suggesting a correlation between neurological symptoms and poor prognosis in SFTS patients [2].

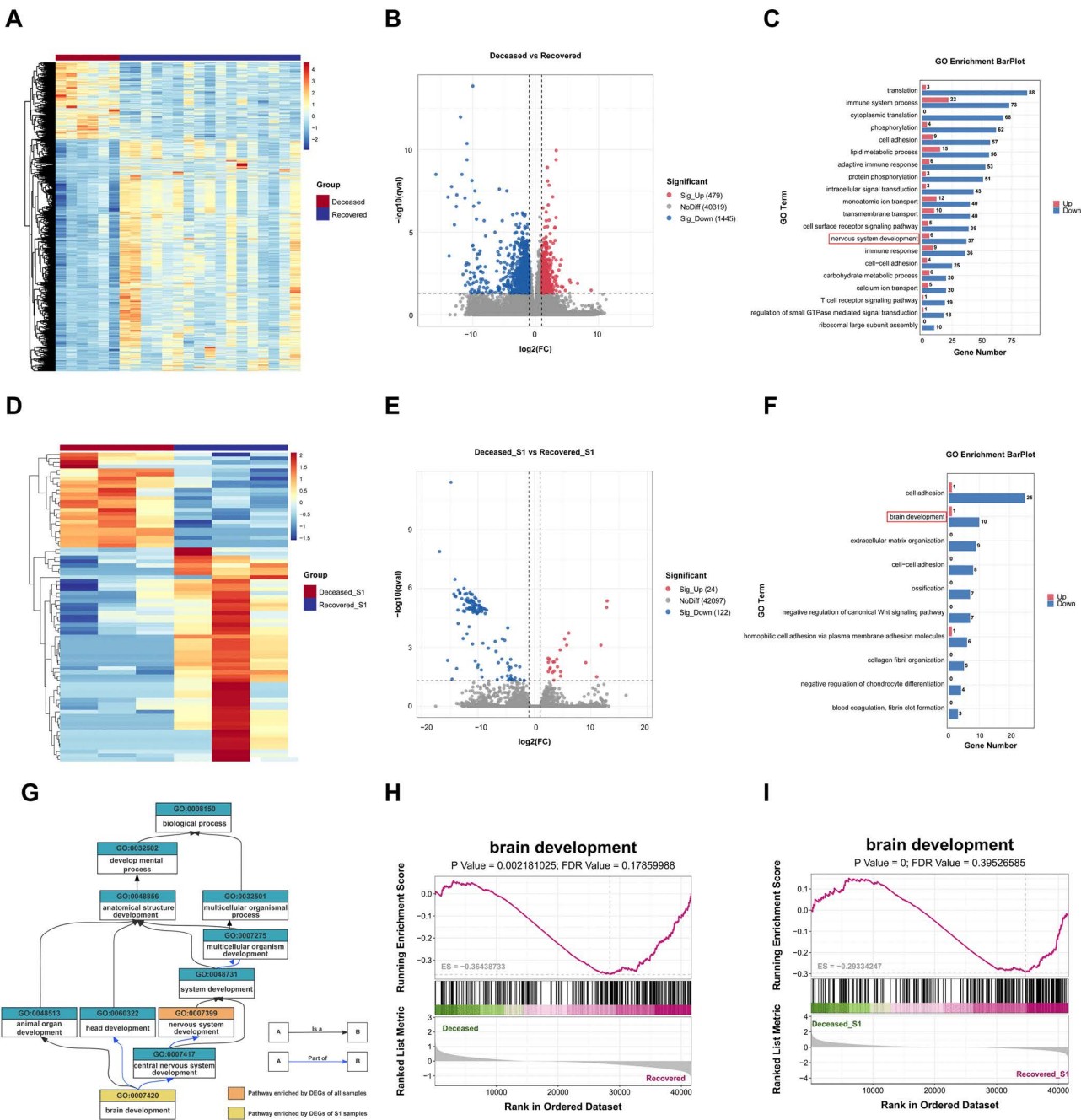

**Fig 2. Transcriptomic differences between the deceased group and the recovered group. A** Heat map of the z-scores for 1924 significantly differentially expressed genes identified using RNA-seq (P < 0.05, exact test, likelihood ratio test, and quasi-likelihood F test) shows that genes distinguish the overall Deceased patients' samples from the Recovered patients' samples. Red represents increased relative expression, and blue represents decreased relative expression. **B** Volcano plot of differential gene expression in overall SFTS deceased group and recovered group with P < 0.05, exact test, likelihood ratio test, and quasi-likelihood F test. This analysis identified 1,445 upregulated (red) and 479 downregulated (blue) genes in the deceased group, highlighting a widespread transcriptomic dysregulation associated with fatal outcomes. FC, fold change; CPM, read count per million. **C** Gene Ontology (GO) enrichment analysis of all DEGs. The bar chart shows the top 20 enriched biological process pathways, with the most significant finding being the enrichment of pathways related to nervous system development in the deceased group. **D** Heat map of the z-scores for 146 DEGs identified using RNA-seq (P < 0.05, exact test, likelihood ratio test, and quasi-likelihood F test) shows that genes distinguish the Deceased patients' samples from the Recovered patients' samples at S1. **E** Volcano plot of DEGs between deceased and SFTS recovered groups at the S1 stage. This identified 146 DEGs, indicating that significant transcriptomic changes predictive of outcome are already present early in the disease course. **F** Top 10

Enriched Biological Process pathways by GO analysis of DEGs between Deceased patients and Recovered patients at S1. This finding is crucial as it shows that early-stage differences are enriched in pathways related to coagulation and brain development, consistent with the clinical progression of severe SFTS. Hierarchical diagram from the GO database illustrating the relationship between the enriched nervous system pathways. This visualizes how "brain development" (identified in the S1 analysis) is a specific sub-pathway of "nervous system development" (identified in the overall analysis). **H** Gene Set Enrichment Analysis (GSEA) plot for the "brain development" pathway using all samples. The negative enrichment score (NES) confirms that genes associated with this pathway are significantly downregulated in the deceased group compared to the recovered group over the entire disease course. **I** GSEA plot for the "brain development" pathway using only S1 stage samples. The result demonstrates that the downregulation of these crucial neural development genes is a very early event, detectable in the initial febrile stage of the illness in patients with a fatal prognosis.

To identify potential early markers of SFTS outcomes, we focused on early-stage samples (S1) with different prognoses. Among the 6 S1 samples, (3 deceased, 3 recovered), differential expression analysis identified 146 DEGs, including 24 genes upregulated and 122 genes downregulated in deceased patients (Fig 2D, 2E and S2 Table). As is shown in Fig 2F, the GO enrichment analysis revealed that coagulation system pathways were regulated, which may contribute to the elevated D-dimer levels (4.68 mg/L FEU vs 1.11 mg/L FEU; P = 0.026) and prolonged APTT values (59.70 ± 16.71s vs 33.43 ± 6.72s; P < 0.001) in the deceased patients (Table 1). Notably, these DEGs were also enriched in the brain development pathway, a subset of the nervous system development pathway (Fig 2G).

Gene Set Enrichment Analysis (GSEA) further confirmed the downregulation of brain development-related genes in deceased patients during the S1 stage (Fig 2H). The enrichment score (ES = -0.3644, P = 0.0022, FDR = 0.1786) underscores the pronounced downregulation of these genes in the deceased group as a whole (Fig 2I). Ranked metric distributions demonstrated the clustering of brain development-related genes towards the deceased group, suggesting their potential involvement in the severe progression of SFTS.

## The association between clinical neurological features and prognosis in SFTS patients

To further explore the relationship between neurological symptoms and clinical outcomes in SFTS patients, detailed neurological symptoms and cerebrospinal fluid (CSF) data were collected for the 15 patients included in the RNA-seq study. Significant differences in neurological symptoms were observed between the recovered and deceased groups. Consciousness disorder was significantly more common in deceased patients compared to recovered patients(100% vs 9.09%, P = 0.004) (Table 2). Similarly, restlessness was markedly associated with deceased patients (75% vs 9.1%, P = 0.033). While other symptoms such as tremor, convulsion, and mental state changes were more frequently observed in deceased patients, these differences did not reach statistical significance (P > 0.05). CSF analysis showed no statistically significant differences between groups. Deceased patients had a lower average CSF pressure (85.00 ± 21.21 mmH₂O) compared to recovered patients (130.00 ± 72.11 mmH₂O), though this difference was not significant (P = 0.472). Similarly, no significant differences were found in karyocyte count, chloride, glucose, or protein levels (P > 0.05).

To investigate whether early neurological symptoms could predict clinical outcomes, we analyzed 6 SFTS patients in the S1 stage for key neurological symptoms. Among S1-stage patients, consciousness disorders were more prevalent in the deceased group (50%) compared to none in the recovered group, with a borderline significant P value of 0.057 (Table 2). Convulsions were observed in 25% of deceased patients but were absent in recovered patients (P = 0.267). These trends align with the findings from the full cohort. Collectively, these results suggest that early neurological symptoms, such as consciousness disorders and convulsions, may serve as critical indicators of poor prognosis in SFTS patients. These findings warrant further investigation to better understand their prognostic potential and guide timely clinical interventions.

## Retrospective validation of neurological symptoms as prognostic indicators in SFTS

In the retrospective cohort (Validation Cohort 1), consciousness disorder emerged as a significant prognostic indicator, with a markedly higher prevalence in the deceased group compared to the recovered group (88.89% vs 23.53%,

**Table 2. Comparison of Neurological Symptoms and Cerebrospinal Fluid (CSF) Parameters of Patients with SFTS between the Recovered and Deceased groups of the Sampling SFTS Patients, Including Subgroup Analysis in the S1 Stage.**

| Neurological Symptoms | Recovered Group (n=11) | Deceased Group (n=4) | Odds Ratio (OR) | P value | S1 | | | |
| --- | --- | --- | --- | --- | --- | --- | --- | --- |
| | | | | | Recovered Group (n=3) | Deceased Group (n=3) | Odds Ratio (OR) | P value |
| Consciousness disorder, n(%) | 1 (9.09) | 4 (100.00) | 0.09 | **0.004** | 0 (0.00) | 2 (50.00) | 0.154 | 0.057 |
| Headache, n(%) | 2 (18.18) | 0 (0.00) | 0.82 | 0.524 | 0 (0.00) | 0 (0.00) | | |
| Restlessness, n(%) | 1 (9.09) | 3 (75.00) | 30.00 | **0.033** | 0 (0.00) | 1 (25.00) | 0.214 | 0.267 |
| Tremor, n(%) | 5 (45.45) | 2 (50.00) | 1.20 | 0.662 | 4 (36.36) | 1 (25.00) | 0.583 | 0.593 |
| Convulsion, n(%) | 1 (9.09) | 1 (25.00) | 3.33 | 0.476 | 0 (0.00) | 1 (25.00) | 0.214 | 0.267 |
| Mental state change, n(%) | 2 (18.18) | 3 (75.00) | 13.50 | 0.077 | 2 (18.18) | 1 (25.00) | 1.500 | 0.637 |
| *CSF Parameter (n=5)* | | | | | | | | |
| Pressure (mmH$_2$O) | 130.00±72.11 | 85.00±21.21 | | 0.472 | | | | |
| Karyocyte count (/µL) | 1.33±1.53 | 2.00±1.41 | | 0.658 | | | | |
| Chloride (mmol/L) | 127.33±7.10 | 123.50±4.95 | | 0.562 | | | | |
| Glucose (mmol/L) | 7.17±2.90 | 4.00±1.13 | | 0.253 | | | | |
| Protein (g/L) | 0.52±0.22 | 0.37±0.06 | | 0.443 | | | | |

P<0.001) (Table 3, Fig 3A), consistent with findings from the prospective cohort (Table 2). Similarly, convulsions were notably more frequent among deceased patients (27.78% vs 3.53%, p=0.003) (Fig 3E), further reinforcing their link to adverse prognoses (Table 3). However, no statistically significant differences emerged in neurological manifestations—including restlessness, headache, tremor, and mental state change—between the deceased and recovered groups (Fig 3B, 3C, 3D, 3F).

While in the analysis of SFTS patients in S1 phase, neurological symptoms including consciousness disorder (72.22% vs 7.06%, P<0.001), restlessness (27.78% vs 3.53%, P=0.003), convulsion (11.11% vs 1.18%, P=0.023) and mental

**Table 3. Comparison of Neurological Symptoms and Cerebrospinal Fluid (CSF) Parameters of Patients with SFTS Between the Recovered and Deceased Groups of the Validation Cohort 1, Including Subgroup Analysis in the S1 Stage.**

| Neurological Symptoms | Recovered Group(n=85) | Deceased Group (n=18) | Odds Ratio (OR) | P value | S1 | | | |
| --- | --- | --- | --- | --- | --- | --- | --- | --- |
| | | | | | Recovered Group (n=85) | Deceased Group (n=18) | Odds Ratio (OR) | P value |
| Consciousness disorder, n(%) | 20 (23.53) | 16 (88.89) | 26.00 | **<0.001** | 6 (7.06) | 13 (72.22) | 34.23 | **<0.001** |
| Headache, n(%) | 9 (10.59) | 2 (11.11) | 1.06 | 0.948 | 4 (4.71) | 1 (5.56) | 1.19 | 0.881 |
| Restlessness, n(%) | 2 (2.35) | 2 (11.11) | 5.19 | 0.082 | 3 (3.53) | 5 (27.78) | 10.51 | **0.003** |
| Tremor, n(%) | 7 (8.24) | 3 (16.67) | 2.23 | 0.304 | 3 (3.53) | 3 (16.67) | 5.47 | 0.058 |
| Convulsion, n(%) | 3 (3.53) | 5 (27.78) | 10.51 | **0.003** | 1 (1.18) | 2 (11.11) | 10.50 | **0.023** |
| Mental state change, n(%) | 8 (9.41) | 1 (6.25) | 0.57 | 0.580 | 13 (15.29) | 16 (88.89) | 44.31 | **<0.001** |
| *CSF Parameter (n=18)* | | | | | | | | |
| Pressure (mmH$_2$O) | 146.00±54.36 | 133.33±11.55 | | 0.702 | | | | |
| Karyocyte count (/µL) | 4.50±3.90 | 3.33±2.08 | | 0.741 | | | | |
| Chloride (mmol/L) | 126.50±6.91 | 130.67±8.96 | | 0.377 | | | | |
| Glucose (mmol/L) | 4.71±1.54 | 5.47±1.01 | | 0.434 | | | | |
| Protein (g/L) | 0.47±0.23 | 0.88±0.45 | | **0.020** | | | | |

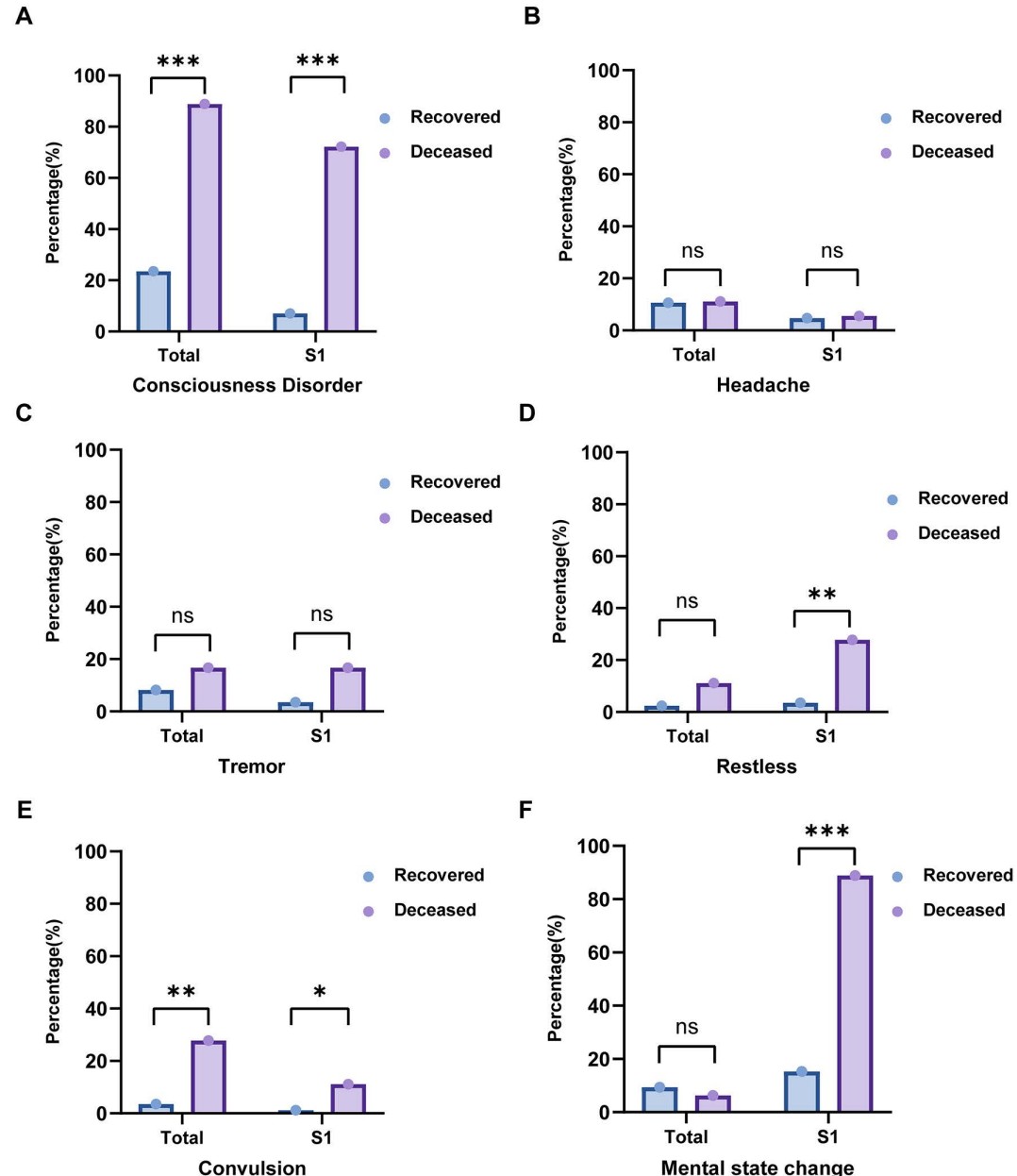

**Fig 3. The Proportion of Different Neurological Symptoms in The Deceased Group and The Recovered Group of the Validation Cohort 1 during S1 and Overall Stage Respectively.** The Vertical Axis Represents the percentage of patients in this group who exhibited certain neurological symptom. A,B,C,D,E,F represents the proportion of patients with consciousness disorder, headache, restlessness, tremor, convulsion and mental state change in all patients and patients in S1 respectively. Statistical significance between the deceased and recovered groups for each symptom was determined using Fisher's exact test.

state change (88.89% vs 15.29%, P < 0.001) were significantly associated with mortality (Fig 3A, 3C, 3E, 3F). These findings highlight their value as potential early indicators of poor prognosis in SFTS patients.

This expanded dataset from the retrospective cohort corroborates the earlier findings, emphasizing the prognostic importance of neurological symptoms, particularly consciousness disorder and convulsion, in predicting outcomes for

SFTS patients. Their consistent association with mortality underscores the need for heightened clinical vigilance and timely interventions for patients presenting with these symptoms.

### *MMP8* as a neurological symptom-associated biomarker predicting mortality in SFTS: Integrated transcriptomic and serum-based validation

Results from transcriptomic analysis and retrospective cohort study suggest a correlation between SFTS and neurological symptoms. To further explore the underlying mechanisms, patients were stratified into neurological symptom-positive (NS) and symptom-negative (no-NS) groups based on the presence of neurological manifestations, specifically altered consciousness and convulsion. Differential gene expression analysis between these cohorts identified 87 significantly differentially expressed genes (adjusted p-value, q < 0.05), comprising 8 upregulated and 79 downregulated genes in the NS group compared to the no-NS group (Fig 4A). Among the 8 upregulated genes, after filtering out low expression genes (mean FPKM <5), we prioritized *MMP8*(matrix metallopeptidase 8) for further scrutiny due to its established roles in inflammatory cascades and extracellular matrix remodeling (Fig 4B) (S2 Table).

We further quantified serum *MMP8* levels through ELISA in 151 hospitalized SFTS patients from two medical centers (43 SFTS patients from The First Affiliated Hospital of Medical School of Zhejiang University and 108 SFTS patients from Nanjing Drum Tower Hospital) and 19 healthy controls to verify the correlation between *MMP8* and neurological symptoms (Table 4).

The results showed that *MMP8* concentrations were significantly elevated only in the patient group with lethal neurological symptoms (consciousness disorder or convulsion). Their levels were significantly higher than both the levels in patients without such symptoms (p = 0.001) and in the healthy control group (p = 0.019). Notably, there was no significant difference in *MMP8* levels between SFTS patients without neurological symptoms and the healthy controls. This finding indicates that *MMP8* elevation is specifically associated with the onset of severe neurological manifestations. This trend was significantly replicated in the larger Validation Cohort 2B (S2B Fig). Although a similar pattern of *MMP8* elevation was observed in Validation Cohort 2A, it did not reach statistical significance, potentially due to the smaller sample size of this subgroup (n = 43) and the consequently limited statistical power (S2A Fig).

To evaluate *MMP8*'s prognostic value, comparative analysis was conducted and revealed significantly elevated *MMP8* concentrations in deceased patients compared to both survivors (p < 0.0001) and healthy controls (p < 0.001) (Fig 4D). Importantly, the *MMP8* levels in the recovered patient group did not differ significantly from those of the healthy control group, suggesting that these levels may normalize with successful recovery. These results were consistent in the independent validation analyses conducted at both centers (S2C, S2D Fig).

To further investigate the specificity of *MMP8* as a biomarker, we also measured serum levels of *MMP9*, another metalloproteinase implicated in blood-brain barrier disruption. In contrast to *MMP8*, our analysis revealed that *MMP9* concentrations were not significantly different between deceased and recovered patients, nor between patients with or without neurological symptoms (S3 Fig).

Cox multivariate analysis identified *MMP8* as an independent predictor of mortality, with a hazard ratio (HR) of 1.060 (95% CI: 1.010-1.112) (S3 Table). Receiver operating characteristic (ROC) curve analysis in our cohort (Validation Cohort 2A, 43 SFTS patients from The First Affiliated Hospital of Medical School of Zhejiang University) demonstrated robust discriminatory performance of *MMP8* for predicting fatal outcomes, with an area under the curve (AUC) of 0.821 (95% CI: 0.681-0.962, p = 0.008) (Fig 4E). The optimal cut-off value was determined as 18.69ng/mL, balancing sensitivity (100.0%, 95% CI: 59.04%-100%) and specificity (61.1%, 95% CI: 43.46%-76.86%).

Consistent with our cohort, the ROC analysis of another independent cohort (Validation Cohort 2B, 108 SFTS patients from Nanjing Drum Tower Hospital) produced an AUC of 0.827 (95%CI: 0.721-0.932, p < 0.001), confirming the stability of *MMP8* as a prognostic biomarker across SFTS patients (Fig 4F). Importantly, when applying the predefined cut-off value of 18.69ng/mL to the Validation Cohort 2B, *MMP8* demonstrated comparable diagnostic performance, with a sensitivity of

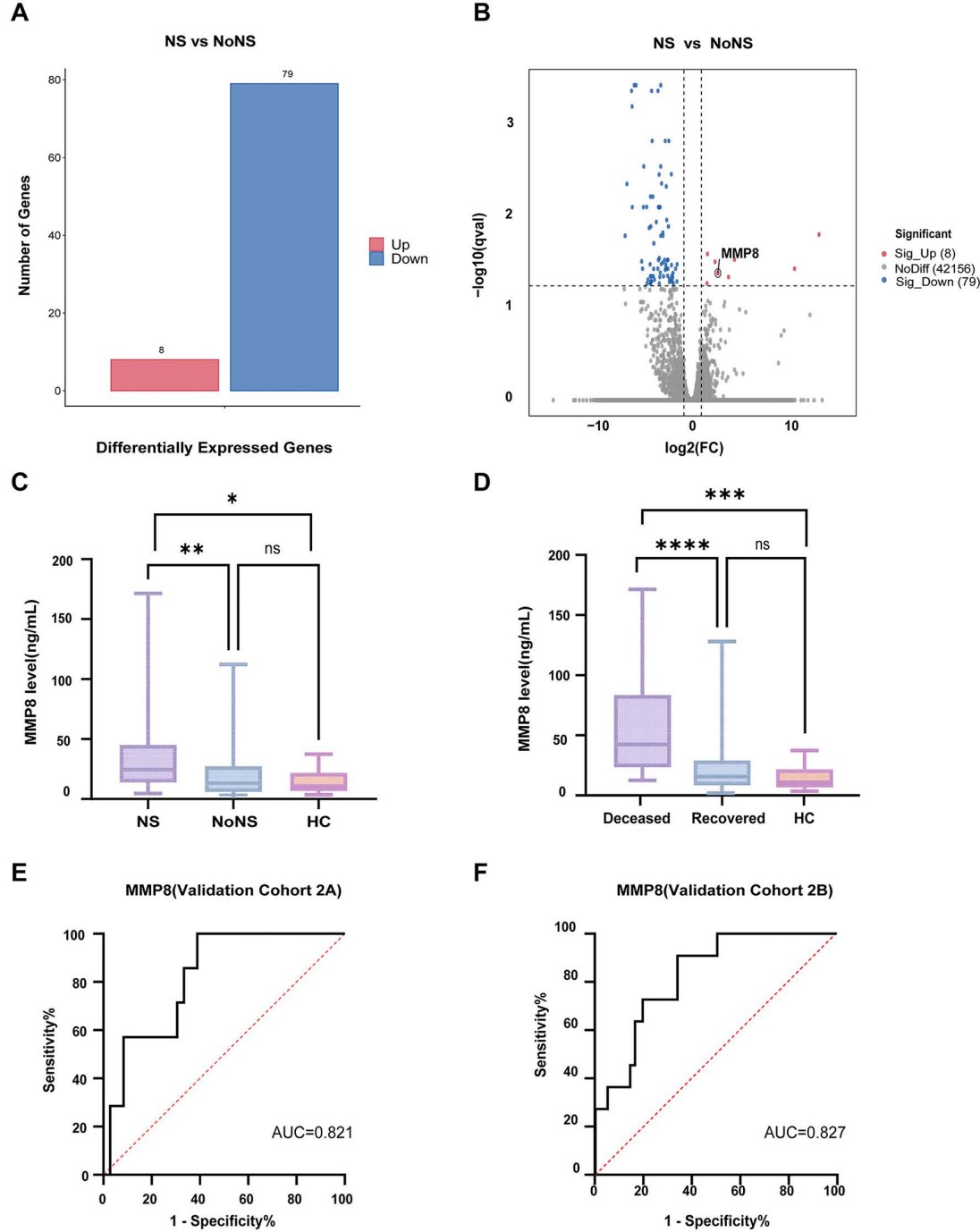

**Fig 4. Variations of *MMP8* levels in gene expression and serum concentrations. A** Histogram of gene expression differences between the neurological symptom-positive (NS) and negative (noNS) groups. Red indicates increased relative expression, and blue indicates decreased relative expression. **B** Volcano plot of differential gene-expression in SFTS patients with neurological symptoms compared to those without. Genes with a q-value < 0.05 are highlighted. Red and blue points indicate significant upregulation or downregulation, respectively, while gray points are not significant. FC, fold change. **C** Comparison of serum *MMP8* concentrations between patient groups with lethal neurological symptoms, those without, and healthy controls. **D** Comparison of serum *MMP8* concentrations among the deceased group, the recovered group, and healthy controls. For (C) and **(D)**, statistical significance across the three groups was assessed using the Kruskal-Wallis test with Dunn's multiple comparisons test. **E** Receiver operating characteristic (ROC) curve demonstrating the predictive ability of *MMP8* for mortality in Validation Cohort 2A. The area under the curve (AUC) and optimal cut-off value are provided. **F** ROC curve analysis of *MMP8* expression for distinguishing non-survivors from survivors in the Validation Cohort 2B.

**Table 4. Baseline Characteristics of SFTS Patients in Validation Cohort 2A and Validation Cohort 2B.**

| Variables | Validation Cohort 2A | | | Validation Cohort 2B | | |
|---|---|---|---|---|---|---|
| | Recovered (n=36) | Deceased (n=7) | *P* value | Recovered (n=97) | Deceased(n=11) | *P* value |
| *Demographics* | | | | | | |
| Age, years | 62.03±11.61 | 72.57±8.56 | **0.028** | 61.60±12.66 | 65.00±14.51 | 0.407 |
| Males,n (%) | 15 (41.67) | 5 (71.43) | 0.303 | 41(42.27) | 4 (36.36) | 0.957 |
| Hypertension,n (%) | 10 (27.78) | 4 (57.14) | 0.282 | 28 (28.87) | 2 (18.18) | 0.693 |
| Diabetes, n (%) | 5 (13.89) | 1 (14.29) | 0.681 | 13 (13.40) | 1(9.09) | 1.000 |
| Tumor, n (%) | 3 (8.33) | 2 (28.57) | 0.180 | 4 (4.12) | 0(0.00) | 0.647 |
| *Laboratory parameters* | | | | | | |
| WBC, ×$10^9$/ L | 4.03(2.76-6.14) | 5.84(4.12-10.30) | 0.199 | 2.80(1.85-4.25) | 2.06(1.90-3.30) | 0.559 |
| Neutrophil (%) | 64.24±15.50 | 76.86±14.15 | 0.053 | 62.31±18.50 | 63.96±12.16 | 0.968 |
| Lymphocyte(%) | 25.32±12.11 | 16.14±10.36 | 0.054 | 28.61±13.62 | 27.58±8.40 | 0.776 |
| RBC, ×$10^{12}$/L | 4.10 (3.79-4.70) | 3.98 (3.75-4.42) | 0.780 | 4.41±0.65 | 4.01±0.85 | 0.215 |
| Hemoglobin, g/L | 124.00 (115.00-139.75) | 128.00 (111.00-145.00) | 0.669 | 133.23±17.66 | 122.09±32.14 | 0.277 |
| Platelet, ×$10^9$/L | 71.00 (48.00-128.00) | 29.00 (18.00-65.00) | **0.012** | 66.00 (41.50-90.50) | 54.00 (31.00-73.00) | 0.284 |
| CRP, mg/L | 3.74(0.80-9.09) | 17.21(6.14-33.34) | **0.035** | 5.85(3.94-9.36) | 7.90(3.44-12.50) | 0.402 |
| PT, s | 11.40 (10.63-12.00) | 12.10 (10.30-13.40) | 0.284 | 11.60 (11.00-12.25) | 11.90 (11.40-12.10) | 0.707 |
| APTT, s | 37.10±9.24 | 75.76±36.18 | **<0.001** | 42.14±24.92 | 50.52±14.31 | **0.014** |
| INR | 0.97(0.91-1.05) | 1.07(0.90-1.18) | 0.161 | 1.03(0.96-1.07) | 1.05(1.00-1.08) | 0.321 |
| D-dimer (mg/L FEU) | 1.31 (0.84-3.38) | 8.35 (4.06-11.39) | **<0.001** | 0.24 (0.08-0.55) | 0.49 (2.82-9.32) | **0.029** |
| ALB (g/L) | 32.88±4.99 | 28.43±3.57 | **0.011** | 37.12±5.60 | 34.17±4.39 | 0.063 |
| ALT (U/L) | 64.00 (27.25-115.25) | 116.00 (49.00-251.00) | 0.130 | 63.00 (39.50-110.45) | 99.00 (58.40-269.00) | 0.086 |
| AST (U/L) | 94.50 (38.00-260.25) | 462.00 (252.00-1421.00) | **0.002** | 128.00 (76.00-234.60) | 397.00 (211.00-549.30) | **0.005** |
| TBIL (µmol/L) | 10.40(6.43-14.93) | 16.00(5.20-19.60) | 0.430 | 9.10(6.50-12.40) | 11.60(9.00-15.70) | 0.103 |
| sCr, µmol/L | 68.06±21.05 | 131.43±44.59 | **<0.001** | 70.00 (57.50-90.50) | 83.00 (70.00-175.00) | **0.021** |
| BUN, mmol/L | 5.74±2.66 | 13.50±5.86 | **<0.001** | 5.20(3.88-8.17) | 7.70(6.07-14.20) | **0.013** |
| LDH (U/L) | 321.50 (258.75-784.75) | 1819.00 (656.00-4465.00) | **0.002** | 647.00 (386.00-1039.50) | 1495.00 (862.00-2077.00) | **0.010** |
| CK (U/L) | 126.50 (54.25-436.25) | 1151.00 (325.00-3192.00) | **0.003** | 264.00 (118.50-641.00) | 1161.00 (296.00-1407.00) | **0.026** |
| Viral Load (TCID$_{50}$/mL) | 220.68 (12.62-2335.41) | 5335632.00 (2534643.00-31315594.00) | **<0.001** | NA | NA | |

Data are median (IQR) or n (%) or mean±standard deviation. P values were calculated by two-sided Mann-Whitney U test, t test or Pearson's chi-squared test according to the normal distribution of data. S1:0–7 days, Fever stage. WBC white blood cell, RBC red blood cell, CRP C-reactive protein, PT prothrombin time, APTT activated partial thromboplastin time, ALT alanine aminotransferase, AST aspartate aminotransferase, TBIL total bilirubin, sCr serum creatinine, BUN blood urea nitrogen, LDH lactate dehydrogenase, CK creatine kinase.

90.90% and specificity of 58.76%. The minimal attenuation in sensitivity (from 100% to 90.90%) and preserved specificity (~60%) across cohorts provide further support for the robustness of this threshold for mortality risk stratification in heterogeneous SFTS patients.

Given the strong association between serum *MMP8* and adverse clinical outcomes, we next sought to explore potential drivers of its upregulation. To investigate whether *MMP8* expression was directly associated with viral replication, we analyzed the correlation between serum *MMP8* concentrations and SFTSV viral load. The analysis revealed no statistically significant correlation between the two variables (Spearman's r=0.233, p=0.133) (S4 Fig). This suggests that the elevation of *MMP8* may be driven by downstream inflammatory pathways rather than directly by the viral burden itself.

## Development and validation of a multi-parameter prognostic model (MCCD) for SFTS mortality prediction

To enhance the prognostic accuracy beyond single biomarkers, we developed a multi-parameter model (MCCD) through Cox proportional hazards regression in the discovery cohort (Center A, n = 43) (S3 Table). Survival time was defined as days from hospital admission to death, with censoring at 30 days for survivors. Multivariable analysis identified four independent mortality predictors: matrix metallopeptidase 8 (MMP8), creatine kinase (CK), creatinine (Cr), and D-dimer. The final prognostic index was expressed as MCCD = 0.058 × MMP8 (ng/mL) + 0.001 × CK (U/L) + 0.041 × Cr (μmol/L) + 0.229 × D-dimer (mg/L FEU), representing a linear predictor of log-hazard ratios. This linear predictor of log-hazard ratios was operationalized via a nomogram (Fig 5) converting biomarker values to point scores (MMP8: 0–110 ng/mL; CK: 0–9,000 U/L; Cr: 0–220 μmol/L; D-dimer: 0–28 mg/L FEU), with total points (0–180) mapping to 30-day mortality probabilities.

External validation in the independent cohort (Center B, n = 108) demonstrated robust performance. Receiver operating characteristic analysis at 30 days yielded an AUC of 0.843 (95% CI: 0.723-0.962), significantly surpassing individual biomarkers including MMP8 (AUC = 0.827, 95% CI: 0.721-0.932), Cr (AUC = 0.713, 95% CI: 0.563-0.864), CK (AUC = 0.705, 95% CI: 0.529-0.881), and D-dimer (AUC = 0.701, 95% CI: 0.549-0.853) (Fig 6A). Calibration at 30 days revealed excellent agreement between predicted and observed survival probabilities, with minimal prediction error (mean absolute error = 0.044) and 90% of errors ≤0.071 across risk strata (Fig 6B). The model satisfied proportional hazards assumptions (Schoenfeld residuals P > 0.05) and maintained discrimination consistency through 30-day follow-up. The MCCD model integrates complementary pathophysiological markers to provide robust mortality risk stratification, with validated discrimination and calibration across independent cohorts.

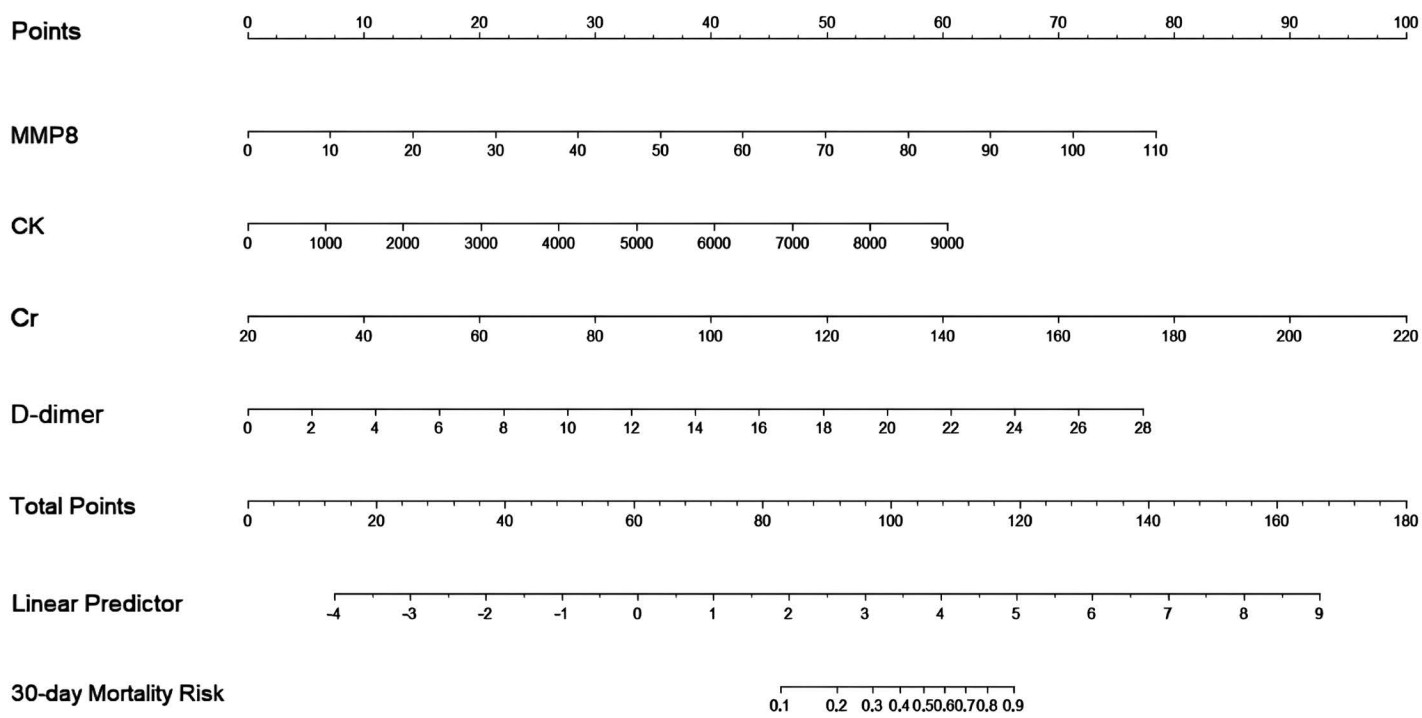

**Fig 5. Nomogram for 30-day mortality risk prediction.** The nomogram assigns points for clinical variables (MMP8, CK, Cr, D-dimer) based on patient values. Points are summed to calculate Total Points (range: 0-180), which correspond to a Linear Predictor value. The Linear Predictor is then converted to predicted 30-day Mortality Risk (range: 0.1-0.9).

## Discussion

Our study elucidates the critical interplay between systemic inflammation, neurological manifestations, and dysregulated gene expression in severe fever with thrombocytopenia syndrome (SFTS), identifying matrix metalloproteinase-8(*MMP8*) as a pivotal prognostic biomarker.

While prior studies have suggested an association between neurological symptoms and mortality in SFTS [25,26], the specific neurological symptoms involved remained indistinct. We addressed this gap by subdividing neurological symptoms into discrete categories: consciousness disorder, headache, restlessness, tremor, convulsion, and mental state change. Crucially, consciousness disorders emerged as a robust predictor of fatal outcomes across both our prospective and retrospective cohorts. Furthermore, analysis within the critical early febrile stage (S1, days 0–7) revealed that specific symptoms like tremor and mental state change also showed differential distributions between outcome groups, potentially reflecting underlying neuropathology even if their association with mortality was less consistent than consciousness disorders. Notably, in the retrospective cohort (Validation Cohort 1, n = 103), the presence of consciousness disorder, restlessness, convulsion, or mental state change during S1 significantly heightened the risk of death. This underscores the imperative for heightened vigilance and intensified therapeutic intervention in SFTS patients exhibiting these particular neurological symptoms in the early disease course. Specifically, such interventions include early consideration for admission to an intensive care unit (ICU) for continuous monitoring and proactive management of key complications like coagulopathy and organ dysfunction. Furthermore, our findings suggest that therapies targeting *MMP8* or protecting the blood-brain barrier warrant investigation in future clinical trials for these high-risk patients.

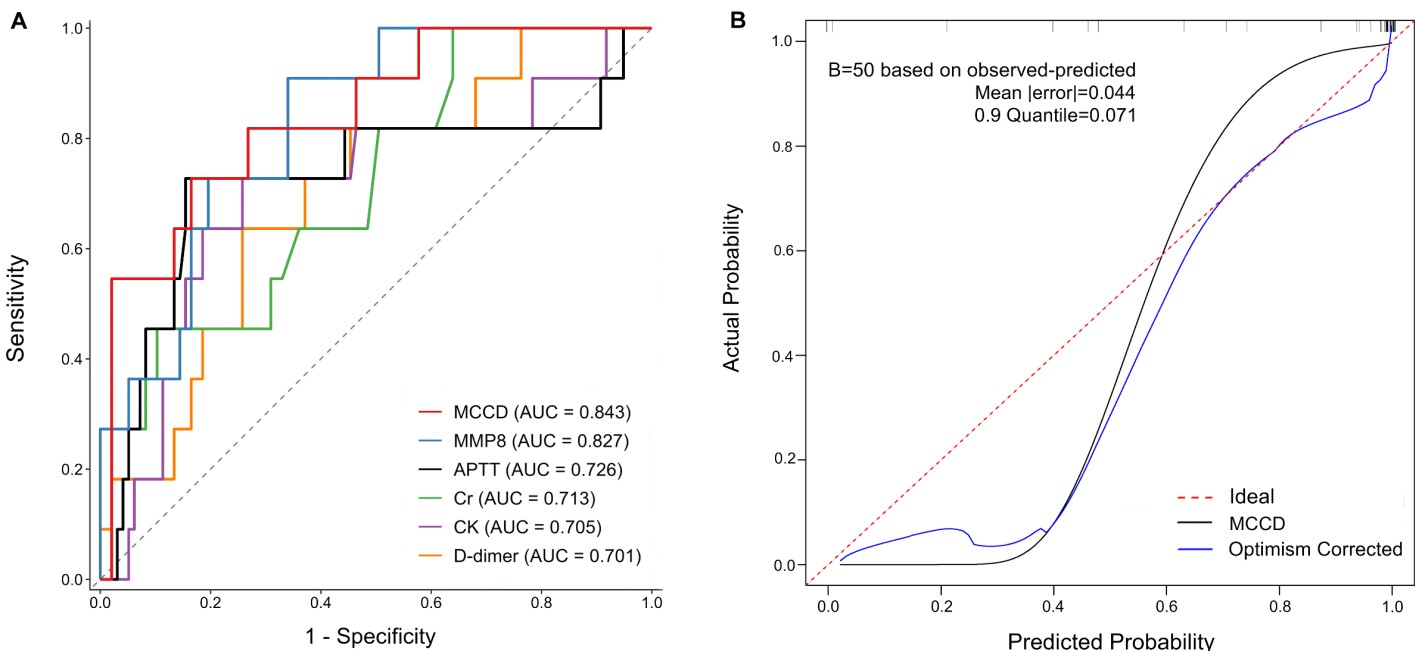

**Fig 6. Model Performance Evaluation: Discrimination and Calibration. A** Receiver operating characteristic (ROC) curves demonstrating the discrimination ability of different prognostic models for 30-day clinical outcomes. The MCCD model (AUC = 0.843) shows superior predictive performance compared to *MMP8* (AUC = 0.827), APTT (AUC = 0.726), creatinine (Cr, AUC = 0.713), creatine kinase-based (CK, AUC = 0.705), and D-dimer (AUC = 0.701) models. **B** Calibration plot assessing agreement between predicted probabilities and observed outcomes. Black stepped line represents actual event rates, gray diagonal denotes perfect calibration, and blue curve shows bias-corrected calibration through bootstrapping (n = 50). Model calibration metrics include a mean absolute error of 0.04 and median (0.5 quantile) absolute error of 0.071, indicating excellent agreement between predicted and observed event probabilities.

Building upon initial transcriptomic profiling that implicated diverse pathways in neurodevelopment and immune regulation, subsequent analysis revealed *MMP8* gene upregulation as a prominent candidate. This focus culminated in serum ELISA validation, establishing *MMP8* as a functionally grounded biomarker. *MMP8*, a member of the matrix metalloproteinase (*MMP*) family, contributes to BBB damage by degrading its structural protein components [27,28]. Its expression is potently upregulated by proinflammatory cytokines (e.g., *TNF-α*, *IL-6*) characteristic of viral CNS infections [29]. Its prominent upregulation in fatal cases and robust discriminatory power (AUC > 0.8) highlight its significant prognostic value.

Reinforcing its role as a marker of the host's immunopathology, our analysis found no direct correlation between serum *MMP8* levels and SFTSV viral load. This finding suggests that *MMP8* upregulation is not driven by the level of viral replication itself but is rather a consequence of the broader, systemic inflammatory cascade that characterizes severe SFTS.

Indeed, the potential of *MMP8* as a biomarker for severity in infectious diseases is an emerging area of research. For example, in pediatric septic shock, *MMP8* levels were found to directly correlate with organ failure and mortality, leading to its proposal as a potential prognostic biomarker for bacterial sepsis [30]. In the context of cerebral malaria, while not explicitly validated as a prognostic marker, its upregulation is considered a key mechanistic factor in the breakdown of the blood-brain barrier [31].

Our study builds upon these findings by validating *MMP8* as an innovative prognostic biomarker for SFTS, where its specific link to nerve injury distinguishes it from conventional laboratory indicators associated with SFTS prognosis, such as APTT, D-dimer, AST, LDH, and Cr [26,32,33]. The limited prognostic value of these routine indicators, combined with the absence of significant differences in standard CSF parameters (e.g., glucose, protein) between outcome groups, underscores a critical advantage of *MMP8*: it provides a peripherally accessible (serum), mechanism-based tool for early risk stratification.

The clinical utility of *MMP8* as a prognostic biomarker is further strengthened by its robust performance across geographically distinct cohorts. Using a predefined cut-off (18.69ng/mL), *MMP8* demonstrated consistent predictive ability. External validation revealed only a modest decrease in sensitivity (from 100% to 90.90%) while maintaining specificity (~60%), strongly suggesting that *MMP8*-driven blood-brain barrier (BBB) disruption represents a conserved, critical feature of fatal SFTS, largely independent of regional or demographic variation. This consistency starkly contrasts with traditional biomarkers like D-dimer or transaminases, which reflect systemic damage but fail to capture CNS injury, a key contributor to fatal outcomes in SFTS. To develop a more comprehensive prognostic tool that integrates neural injury with systemic pathology, we constructed a multi-parameter model (designated MCCD: **MMP8**, **C**K, **C**r, **D**-dimer). The MCCD model incorporates the neural injury biomarker *MMP8* alongside established indicators of major systemic complications: creatine kinase (CK, indicating muscle injury/ myonecrosis), creatinine (Cr, reflecting renal dysfunction/impaired clearance), and D-dimer (denoting coagulopathy/ hypercoagulability). By leveraging these complementary pathophysiological pathways, the MCCD model demonstrated superior discriminative power (AUC = 0.843; 95% CI: 0.723-0.962) compared to any individual biomarker alone during external validation.

Despite these strengths, limitations warrant consideration. First, the study cohort consisted exclusively of Chinese patients, which potentially limits the generalizability of our findings. Validation in other SFTS-endemic populations, such as those in Japan and South Korea, is therefore essential. Second, a primary limitation is the small cohort size for the initial PBMC transcriptomic analysis, which constrained its statistical power. Therefore, we appropriately treated this transcriptomic analysis as an exploratory, hypothesis-generating phase of our research. Our confidence in the primary finding is substantially bolstered by their successful validation in larger, independent cohorts (n = 103 and n = 151), which confirmed the prognostic significance of neurological symptoms and serum *MMP8* levels. Another limitation is the lack of dynamic *MMP8* analysis in our main cohorts. To preliminarily explore this, we analyzed longitudinal samples from four patients (3 recovered and 1 deceased) (S5 Fig). Despite the small sample size, a clear distinction emerged: the single fatal case showed a progressively rising trajectory, whereas the three recovered patients showed variable trends. Additionally, the absence of cerebrospinal fluid (CSF) samples, precluded the direct measurement of CSF *MMP8* levels. However, prior evidence suggests that serum *MMP8* exhibits stronger predictive value for blood-brain barrier (BBB) disruption than CSF *MMP8* in tickborne encephalitis (TBE) patients [28]. Therefore, future prospective studies with both serial serum and CSF sampling are warranted to validate these dynamic patterns and directly investigate the link between systemic and CNS *MMP8* expression. Mechanistic studies in animal models are also essential to dissect *MMP8*'s contributions to neuroinflammation.

In conclusion, this study establishes *MMP8* as a critical prognostic biomarker in SFTS, mechanistically linking systemic inflammation and cytokine storms to neurological complications and BBB disruption. Its association with specific, prognostically significant neurological symptoms (especially early consciousness disorders), robust discriminatory power, and validation across cohorts highlight its clinical utility for early risk stratification. This multi-faceted approach offers significant clinical advantages: clinicians can use neurological symptoms for immediate, non-invasive risk stratification at the bedside. Furthermore, *MMP8* as a standalone biomarker demonstrates superior prognostic performance over traditional indicators like APTT and D-dimer; and the *MMP8*-inclusive MCCD model provides the highest overall predictive accuracy. These findings underscore the importance of detailed neurological symptom profiling and mechanism-driven biomarkers like *MMP8*, paving the way for improved patient management and personalized therapeutic strategies. Future research must prioritize validating *MMP8* in larger cohorts, elucidating its precise role in SFTS neuropathogenesis through mechanistic studies, and exploring its potential as a therapeutic target to mitigate neurological complications and improve outcomes.

## Supporting information

**S1 Table. Differential expression of genes between deceased group and recovered group during S1.**
(XLSX)

**S2 Table. The expression of 8 up-regulated genes of the group with neurological symptoms compared to the group without neurological symptoms.**
(XLSX)

**S3 Table. Risk factors associated with 30-day mortality of SFTS patients by univariate and multivariate cox regression analysis.**
(DOCX)

**S1 Fig. Transcriptomic re-analysis of deceased versus recovered SFTS patients using a stringent FDR-adjusted statistical threshold (q < 0.05).** (A) Hierarchical clustering heatmap of 1,693 differentially expressed genes (DEGs) identified with a q-value<0.05. The heatmap shows a distinct gene expression signature that clearly separates deceased from recovered patient samples. (B) Volcano plot identifying 611 upregulated (red) and 1,082 downregulated (blue) genes in the deceased group. (C) Gene Ontology (GO) enrichment analysis of the DEGs, confirming that nervous system development pathways remain significantly enriched.
(TIF)

**S2 Fig. Association of serum *MMP8* levels with neurological symptoms and mortality in independent validation cohorts.** (A-B) Comparison of serum *MMP8* concentrations among SFTS patients with neurological symptoms (NS), without neurological symptoms (noNS), and healthy controls in Validation Cohort 2A (A) and Cohort 2B (B). (C-D) Comparison of serum *MMP8* levels among deceased patients, recovered patients, and healthy controls in Validation Cohort 2A (C) and Cohort 2B (D). Boxplots show the median, interquartile range, and full data range. Statistical significance across the three groups in each comparison was determined using the Kruskal-Wallis test with Dunn's multiple comparisons test. Significant differences are indicated (*$P < 0.05$, **$P < 0.01$, ***$P < 0.001$, ****$P < 0.0001$).
(TIF)

**S3 Fig. Serum *MMP9* concentrations are not associated with clinical outcomes or neurological symptoms in SFTS patients.** (A) Comparison of serum *MMP9* concentrations among the deceased group and the recovered patient group. (B) Comparison of serum *MMP9* concentrations between patient groups with and without lethal neurological symptoms (consciousness disorder or convulsion). No statistically significant differences were observed between the patient groups in either analysis (Mann-Whitney U test).
(TIF)

**S4 Fig. Correlation between serum *MMP8* levels and viral load.** No significant correlation was observed between serum *MMP8* (Y-axis) and viral load (X-axis) in SFTS patients (Spearman's correlation, $P=0.133$).
(TIF)

**S5 Fig. Exploratory analysis of longitudinal serum *MMP8* dynamics in a subset of discovery cohort patients.** The line graph depicts the serum *MMP8* concentration trajectories across different disease stages (S1, S2, S3) for four patients: one who ultimately died and three who recovered. A clear distinction in the patterns emerged: the fatal case showed a progressively rising trajectory, whereas the recovered patients displayed variable trends. This analysis is presented as preliminary and descriptive due to the small sample size; no formal statistical tests were performed.
(TIF)

## Author contributions

**Conceptualization:** Jie Wang, Wei Wu.

**Data curation:** Qi Xia, Ziling Cheng, Bei Jia, Wei Wu.

**Formal analysis:** Qi Xia, Ziling Cheng.

**Funding acquisition:** Qi Xia, Wei Wu.

**Methodology:** Lingtong Huang, Jie Wang, Wei Wu.

**Project administration:** Qi Xia, Jie Wang, Wei Wu.

**Resources:** Qi Xia, Ziling Cheng, Yi Zhang, Haolin Song, Qiuhong Liu, Qing Zhao, Jie Li.

**Writing – original draft:** Qi Xia, Ziling Cheng, Jie Wang.

**Writing – review & editing:** Jie Li, Jie Wang, Wei Wu.

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
