## [Decision Letter · Decision Letter 0]

18 Sep 2025

Neurological manifestations and MMP8 as a prognostic biomarker in Severe Fever with Thrombocytopenia Syndrome

Dear Dr. Wu,

Thank you for submitting your manuscript to PLOS Neglected Tropical Diseases. After careful consideration, we feel that it has merit but does not fully meet PLOS Neglected Tropical Diseases's publication criteria as it currently stands. Therefore, we invite you to submit a revised version of the manuscript that addresses the points raised during the review process.

Please submit your revised manuscript within 60 days Nov 17 2025 11:59PM. If you will need more time than this to complete your revisions, please reply to this message or contact the journal office at plosntds@plos.org. Please include the following items when submitting your revised manuscript:

We look forward to receiving your revised manuscript.

Kind regards,

Shu Shen

Academic Editor

Michael Holbrook

Section Editor

Shaden Kamhawi

co-Editor-in-Chief

Paul Brindley

co-Editor-in-Chief

**Additional Editor Comments:**

Reviewer #1:

Reviewer #2:

Reviewer #3:

Reviewer #4:

**Journal Requirements:**

1) Please amend your detailed Financial Disclosure statement. This is published with the article. It must therefore be completed in full sentences and contain the exact wording you wish to be published.

2) Kindly revise your competing statement to align with the journal's style guidelines: 'The authors declare that there are no competing interests.'

**Reviewers' Comments:**

Reviewer's Responses to Questions

**Key Review Criteria Required for Acceptance?**

**Methods**

-Are the objectives of the study clearly articulated with a clear testable hypothesis stated?

-Is the study design appropriate to address the stated objectives?

-Is the population clearly described and appropriate for the hypothesis being tested?

-Is the sample size sufficient to ensure adequate power to address the hypothesis being tested?

-Were correct statistical analysis used to support conclusions?

-Are there concerns about ethical or regulatory requirements being met?

Reviewer #1: (No Response)

Reviewer #2: (No Response)

Reviewer #3: (No Response)

Reviewer #4: Comment1

The manuscript mentions neurological symptoms such as “consciousness disorder,” “restlessness,” and “mental state change.” It is recommended to provide the specific assessment criteria for these symptoms in the Methods section. For example, was “consciousness disorder” quantified using standardized tools such as the Glasgow Coma Scale (GCS)? This is crucial for ensuring the objectivity and reproducibility of the results.

Comment2:

Sample collection time points: For the RNA-seq samples, it was stated that they were collected during hospitalization. Could more specific information be provided—for instance, were all samples collected within 24 hours of admission, or were they collected at different stages of the disease (e.g., S1, S2)? This detail is important for interpreting the dynamic changes in gene expression.

Comment3:

In the Cox regression analysis, how were the candidate variables selected for inclusion in the multivariable model? Were they based on variables with p < 0.05 in the univariable analysis, or determined by clinical expertise? It is suggested to clarify this point.

**Results**

-Does the analysis presented match the analysis plan?

-Are the results clearly and completely presented?

-Are the figures (Tables, Images) of sufficient quality for clarity?

Reviewer #1: (No Response)

Reviewer #2: (No Response)

Reviewer #3: (No Response)

Reviewer #4: comment1

Figure 2 presents a large amount of transcriptomic data. It is recommended that the figure legend provide more detailed descriptions of the purpose and main findings of each subpanel (A–I) to help readers better understand the results.

Comment2:

In the description of the MCCD model, the formula MCCD = 0.058 × MMP8 + 0.001 × CK + 0.041 × Cr + 0.229 × D-dimer is provided. What are the units of each variable? For example, is MMP8 expressed in ng/mL, CK in U/L, etc.? It is suggested to specify the units either alongside the formula or in the main text to facilitate reader interpretation and application.

**Conclusions**

-Are the conclusions supported by the data presented?

-Are the limitations of analysis clearly described?

-Do the authors discuss how these data can be helpful to advance our understanding of the topic under study?

-Is public health relevance addressed?

Reviewer #1: (No Response)

Reviewer #2: (No Response)

Reviewer #3: (No Response)

Reviewer #4: Comment1:

The discussion section could further elaborate on the specific clinical implications of the study findings. For instance, when clinicians detect elevated MMP8 levels in SFTS patients (e.g., exceeding 18.69 ng/mL) accompanied by early signs of altered consciousness, what specific “intensified therapeutic interventions” (line 440) should be implemented? For example, should immediate transfer to the ICU be recommended, should closer monitoring of vital signs be performed, or should certain experimental therapies targeting inflammation or blood–brain barrier protection be considered? Providing more concrete recommendations would greatly enhance the translational value of the manuscript.

Comment2:

The authors mention that future prospective studies are needed to determine whether the peak of MMP8 occurs during the S1 stage. Based on the existing data, some preliminary inferences could be made. For example, among the 15 patients in the discovery cohort, are there multi-time-point samples that could provide an initial depiction of the dynamic trend of serum MMP8 changes? Even if the data are limited, such findings could serve as preliminary evidence and be included in the discussion.

**Summary and General Comments**

Reviewer #1: This study aims to explore relevant mechanisms and identify biomarkers of SFTS. This study presents major shortcomings in clinical relevance, research logic, and experimental design. Firstly, the central research question fails to address a real clinical need, and the identified biomarker lacks adequate comparative analysis and validation. Secondly, the study design suffers from issues such as inappropriate sample selection, unclear sampling time points, and a disconnect between experimental strategies and research objectives. These factors collectively undermine the reliability of the findings and limit their translational value in clinical practice.

1. Over the past several years, numerous studies have explored biomarkers associated with severe SFTS. In fact, many routine hematological parameters—such as APTT and D-dimer, as mentioned in the current manuscript—are already useful in assessing disease progression. Clinicians can often extract meaningful diagnostic clues from standard biochemical monitoring. This raises the question: is there a genuine need to search for additional biomarkers? Unless such biomarkers can effectively distinguish SFTS from other thrombocytopenia-related diseases or enable early identification in the absence of accessible nucleic acid testing, their clinical value may be limited. Therefore, I believe this study does not sufficiently address a practical clinical problem.

2. If the goal is to identify biomarkers predictive of disease progression, then the study design should clearly define the timing of sample collection relative to disease onset. Although the authors mention different disease phases in the Methods section, they fail to specify which phase their samples correspond to or whether sample collection was standardized. Furthermore, PBMCs may not be the most appropriate sample type for this purpose. For translational biomarker discovery, serum samples would be more suitable. Using PBMCs to identify gene expression changes may not yield clinically actionable results. As such, the study's rationale appears disorganized and its objectives unclear.

3. The identification of MMP8 as a biomarker appears insufficiently substantiated. In biomarker research, it is common to propose several candidate molecules and validate them in an independent cohort. However, this study does not report any additional candidates or describe a comprehensive validation process. Even if such work was performed, a rigorous study should include those results. Additionally, neurological symptoms were not described in cohort 1, yet this cohort was used for screening biomarkers associated with neurological manifestations. This discrepancy raises concerns about the appropriateness of the cohort design and the interpretation of findings.

Reviewer #2: In this manuscript, through prospective and retrospective cohort studies, the authors demonstrated that neurological symptoms during the early stages of SFTS might serve as a predictor of fatal outcomes, and also revealed MMP8 as a potential biomarker for SFTS prognosis.

Specific comments:

1. Figure 4C, 4D and Fig S1, the authors detected and compared the MMP8 levels in the serum of SFTS patients. A comparison with healthy controls should also be included.

2. Please analyze the correlation between viral load and MMP8 levels in the serum.

3. Has MMP8 been considered as biomarker in other infectious diseases? The authors should discuss this.

4. Figure legends for Figures 3, 4 and Fig S1 must include descriptions of the statistical methods employed.

Reviewer #3: The objectives of this study are clearly articulated. The authors sought to (i) evaluate the prognostic significance of neurological symptoms in SFTS patients, (ii) explore underlying molecular mechanisms using transcriptomics, and (iii) validate MMP8 as a prognostic biomarker in multiple independent cohorts.

While an explicit hypothesis statement (“We hypothesize that neurological symptoms in SFTS patients are linked to transcriptomic alterations and that MMP8 is an independent prognostic biomarker”) could be more clearly stated, the research question is inherently testable and consistently pursued throughout the manuscript. I have following comments:

1. Diagnosis followed standardized Chinese national guidelines, and inclusion/exclusion criteria were appropriate. The demographic description (age, comorbidities, baseline labs) is well detailed.

The population aligns well with the hypothesis (prognostic biomarkers in SFTS). However, generalizability outside Chinese populations (e.g., Japan, South Korea) remains untested and should be acknowledged.

2. RNA-seq cohort (n=15, 4 deaths): Too small to provide robust statistical power for high-dimensional transcriptomic analysis. This limits confidence in differentially expressed gene calls and enrichment results. Authors should highlight this limitation more strongly.

3. Small RNA-seq sample size may produce false positives. Authors should consider FDR-adjusted thresholds in DEGs instead of relying solely on p<0.05

4. Minor text edits: Correct typographical errors (“datas” → “data”)

5. Since neurological symptoms are central to the study, absence of cerebrospinal fluid MMP8 (or other neuroinflammation markers) is a missed opportunity, if samples still available, it would be helpful to add this data too.

6. Author should also add the recent reports on the presence of this virus from Kenya and Pakistan

7. Dabie bandavirus (DBV) is the new name of SFTSV, author should mention like this while mentioning the old name.

8. Also Bunyaviridae family has been updated to Bunyavirale order, so author should check the recent ICTV classification to update in the manuscript.

Reviewer #4: The manuscript presents a well-designed and comprehensive study with substantial clinical significance. By employing a multicenter, multi-cohort design, the authors systematically examined the relationship between neurological symptoms and adverse outcomes in patients with SFTS. Importantly, the study successfully identified and validated matrix metalloproteinase-8 (MMP8) as a potential prognostic biomarker associated with neurological injury. The research follows a clear and coherent logic, progressing from clinical observations (neurological symptoms), to mechanistic exploration using transcriptomics, back to large-scale clinical validation with serum samples, and finally culminating in the development and validation of a multi-parameter prognostic model (MCCD).

Overall, the manuscript is of high scientific value, supported by robust data, and presents credible conclusions. Nevertheless, certain aspects would benefit from further clarification and refinement, as detailed below.

PLOS authors have the option to publish the peer review history of their article (what does this mean? ). If published, this will include your full peer review and any attached files.

**Do you want your identity to be public for this peer review?** For information about this choice, including consent withdrawal, please see our Privacy Policy .

Reviewer #1: No

Reviewer #2: No

Reviewer #3: No

Reviewer #4: No

**Figure resubmission:**
---

## [Decision Letter · Decision Letter 1]

18 Nov 2025

Response to Reviewers
Revised Manuscript with Track Changes
Manuscript

Shaden Kamhawi

co-Editor-in-Chief

Paul Brindley

co-Editor-in-Chief

**Journal Requirements:**

1) Thank you for including an Ethics Statement for your study. Please state whether the consent obtained is verbal or written.

2) We noted that the data under accession HRA010122 will be available on 2026-05-01. We strongly recommend all authors deposit their data before acceptance, as the process can be lengthy and hold up publication timelines. Please note that, though access restrictions are acceptable now, your entire minimal dataset will need to be made freely accessible if your manuscript is accepted for publication. This policy applies to all data except where public deposition would breach compliance with the protocol approved by your research ethics board. If you are unable to adhere to our open data policy, please kindly revise your statement to explain your reasoning and we will seek the editor's input on an exemption.

**Reviewers' comments:**

**Editorial and Data Presentation Modifications?**

Reviewer #3: Overall the authors have improved the manuscript and I would recommend the paper for the acceptance.

**Summary and General Comments**

Reviewer #2: 1. The quality of the figures needs to be optimized, and the annotations should be standardized, especially in Fig.4, Fig. 6 and Fig. S2.

2. Fig.S2, the '50' in TICD50 should be subscripted.

**Figure resubmission:**

**Reproducibility** To enhance the reproducibility of your results, we recommend that authors of applicable studies deposit laboratory protocols in protocols.io, where a protocol can be assigned its own identifier (DOI) such that it can be cited independently in the future. Additionally, PLOS ONE offers an option to publish peer-reviewed clinical study protocols. Read more information on sharing protocols at https://plos.org/protocols?utm_medium=editorial-email&utm_source=authorletters&utm_campaign=protocols

---

## [Decision Letter · Decision Letter 2]

18 Dec 2025

Dear Dr. Wu,

We are pleased to inform you that your manuscript 'Neurological manifestations and MMP8 as a prognostic biomarker in Severe Fever with Thrombocytopenia Syndrome' has been provisionally accepted for publication in PLOS Neglected Tropical Diseases.

Best regards,

Shu Shen

Academic Editor

Mabel Carabali

Section Editor

Shaden Kamhawi

co-Editor-in-Chief

Paul Brindley

co-Editor-in-Chief

---

## [Editor Report · Acceptance letter]

Dear Dr. Wu,

We are delighted to inform you that your manuscript, " 

Neurological manifestations and MMP8 as a prognostic biomarker in Severe Fever with Thrombocytopenia Syndrome," has been formally accepted for publication in PLOS Neglected Tropical Diseases.

Best regards,

Shaden Kamhawi

co-Editor-in-Chief

Paul Brindley

co-Editor-in-Chief
